# Association between Antiviral Prophylaxis and Cytomegalovirus and Epstein–Barr Virus DNAemia in Pediatric Recipients of Allogeneic Hematopoietic Stem Cell Transplant

**DOI:** 10.3390/vaccines9060610

**Published:** 2021-06-07

**Authors:** Ndeye Soukeyna Diop, Pascal Roland Enok Bonong, Chantal Buteau, Michel Duval, Jacques Lacroix, Louise Laporte, Marisa Tucci, Nancy Robitaille, Philip C. Spinella, Geoffrey Cuvelier, Suzanne M Vercauteren, Victor Lewis, Caroline Alfieri, Helen Trottier

**Affiliations:** 1CHU Sainte-Justine Research Centre, Department of Social and Preventive Medicine, Université de Montréal, Montreal, QC H3T 1C5, Canada; Soukeyna.diop@recherche-ste-justine.qc.ca (N.S.D.); pascal.roland.enok.bonong@umontreal.ca (P.R.E.B.); 2CHU Sainte-Justine Research Centre, Department of Pediatrics, Division of Infectious Diseases, Université de Montréal, Montreal, QC H3T 1C5, Canada; chantal.buteau.hsj@ssss.gouv.qc.ca; 3Department of Pediatrics, Division of Hematology-Oncology, CHU Sainte-Justine, Université de Montréal, Montreal, QC H3T 1C5, Canada; michel.duval@umontreal.ca; 4Department of Pediatrics, Division of Pediatric Intensive Care Medicine, CHU Sainte-Justine, Université de Montréal, Montreal, QC H3T 1C5, Canada; jacques.lacroix.med@ssss.gouv.qc.ca (J.L.); marisa.tucci.med@ssss.gouv.qc.ca (M.T.); 5CHU Sainte-Justine Research Centre, Montreal, QC H3T 1C5, Canada; louise.laporte.hsj@ssss.gouv.qc.ca; 6Department of Pediatrics, Division of Hematology-Oncology, CHU Sainte-Justine, Université de Montréal, and Medical Affairs, Transfusion Medicine, Héma-Québec, Ville Saint-Laurent, QC H4R 2W7, Canada; nancy.robitaille@umontreal.ca; 7St. Louis Children’s Hospital, Washington University School of Medicine, St. Louis, MO 63110, USA; pspinella@wustl.edu; 8Department of Pediatrics and Child Health, CancerCare Manitoba, University of Manitoba, Winnipeg, MB R3E 0V9, Canada; gcuvelier@cancercare.mb.ca; 9Department of Pathology and Laboratory Medicine, BC Children’s Hospital, University of British Colombia, Vancouver, BC V6H 3N1, Canada; svercauteren2@cw.bc.ca; 10Department of Pediatrics and Department of Oncology, Alberta Children’s Hospital, University of Calgary, Calgary, AB T3B 6A8, Canada; Victor.Lewis@albertahealthservices.ca; 11CHU Sainte-Justine Research Centre, Department of Microbiology, Infectiology and Immunology, Université de Montréal, Montréal, QC H3T 1C5, Canada; carolina.alfieri@umontreal.ca

**Keywords:** antiviral prophylaxis, hematopoietic stem cell transplantation, Epstein–Barr virus, cytomegalovirus, human herpesvirus, pediatric

## Abstract

Background: Epstein–Barr virus (EBV) and cytomegalovirus (CMV) infections can have serious consequences during the period of aplasia and lymphopenia following hematopoietic stem cell transplantation (HSCT). Large pediatric cohort studies examining the effect of antiviral prophylaxis against these viruses are scarce. The present study aimed to analyse the potential effect of antiviral prophylaxis (acyclovir and famciclovir) on active post-transplant EBV and CMV infection in a pediatric cohort of allogeneic HSCT recipients. Methods: We used data from the TREASuRE cohort, consisting of 156 patients who had a first allogeneic HSCT, enrolled in four pediatric centers in Canada between July 2013 and March 2017. Follow-up was performed from the time of transplant up to 100 days post-transplant. Adjusted hazard ratio (HR) with 95% confidence intervals (CI) for the association between antiviral prophylaxis with acyclovir and/or famciclovir and EBV and CMV DNAemia was estimated using multivariate Cox regression models. Results: The post-transplant cumulative incidence of EBV and CMV DNAemia at 100 days of follow-up were, respectively, 34.5% (95% CI: 27.6–42.6) and 19.9% (95% CI: 14.5–27.1). For acyclovir, the adjusted hazard ratio (HR) for CMV and EBV DNAemia was 0.55 (95% CI: 0.24–1.26) and 1.41 (95% CI: 0.63–3.14), respectively. For famciclovir, the adjusted HR were 0.82 (95% CI: 0.30–2.29) and 0.79 (95% CI: 0.36–1.72) for CMV and EBV DNAemia, respectively. Conclusion: The antivirals famciclovir and acyclovir did not reduce the risk of post-transplant CMV and EBV DNAemia among HSCT recipients in our pediatric population.

## 1. Introduction

Hematopoietic stem cell transplantation (HSCT) is used for the treatment of malignant tumors and certain blood or immune system disorders [1,2,3,4]. The immunosuppressed state correlates with a significant risk of post-transplant infections, and these infections play a major role in treatment-related morbidity and post-transplant mortality in pediatric HSCT [5,6,7,8,9,10]. Among the most morbid viral infections are those caused by the human herpesviruses (HHV), namely herpes simplex viruses (HSV or HHV-1/2), varicella-zoster virus (VZV or HHV-3), Epstein–Barr virus (EBV or HHV-4), and cytomegalovirus (CMV or HHV-5) [11,12]. Active infection with HHV-6 (roseolavirus) and HHV-7 can also occur but is less common [1,11,12,13,14]. HHV-8 infection (associated with Kaposi’s sarcoma and Castleman’s disease) is infrequent in pediatric HSCT [15,16]. Active HHV infections can occur either during primary infection or following reactivation of latent virus [8], predominantly during the early post-transplant period when the patient’s cell-mediated immune response is severely compromised [9,10,11]. More than two-thirds of patients develop viral reactivation or primary HHV infections in the first 3 months after HSCT [12].

Antivirals are generally used systematically in clinical protocols to prevent reactivation of specific human herpesviruses, namely HSV, VZV, and CMV [17,18,19]. The Francophone Society of Bone Marrow Transplantation and Cell Therapy, and the European Conference on Leukemia recommend prophylaxis for HSV seropositive patients from the start of conditioning until the end of granulocytopenia and resolution of mucositis. Specific recommendations involve the use of intravenous acyclovir or famciclovir for 3 to 6 weeks after initiation of chemotherapy or grafting, or up to the end of aplasia [18,20]. Long-term prophylaxis with acyclovir (up to one-year post-transplant or until immunosuppressive drugs are discontinued) is also recommended for VZV seropositive HSCT recipients. Ganciclovir is recommended until day 100 post-HSCT to prevent active CMV infection in allogeneic HSCT recipients who are at risk (CMV seropositive recipient or CMV seropositive donor) of engraftment failure [21,22]. Although active EBV infection is very common and may lead to serious complications such as post-transplant lymphoproliferative disorder (PTLD), the International Medical Societies do not recommend the use of antiviral prophylaxis for EBV as there is no proven clinically effective antiviral against this virus [23].

Viral blood monitoring can also be done for early detection of viral DNA and pre-emptive treatment. In this case, valaciclovir and famciclovir are preferred for the oral treatment of HSV and VZV in pediatric HSCT recipients with stable localized disease [18]. Intravenous ganciclovir can also be used for the treatment of CMV disease during the year following HSCT [21,22]. Although there are no recognized clinically effective antivirals against EBV, qPCR monitoring is usually performed to assess pre-emptive treatment of patients showing significant spikes in viral load with the anti-CD20 monoclonal (rituximab) thereby preventing the development of PTLD [18].

Overall, the prophylactic efficacy of acyclovir and famciclovir for HSV and VZV is well recognized as a standard-of-care in the clinical guidelines of various transplant societies. However, questions remain about the potential effect of these antivirals on other HHV, such as EBV and CMV. The pediatric literature on this issue is especially limited. The main goal of this study was, therefore, to measure the association between antiviral prophylaxis with acyclovir and famciclovir and post-transplant EBV and CMV DNAemia among pediatric recipients of allogeneic HSCT.

## 2. Methods

### 2.1. Study Design and Participants

We used data from TREASuRE study, a multicenter prospective cohort study of allogeneic HSCT recipients from four Canadian pediatric tertiary care centers. Study design and methods have been published previously [24]. Briefly, the study enrolled 156 patients under the age of 21 who had undergone allogeneic HSCT (bone marrow, cord blood, or peripheral blood). Recruitment was conducted at different sites: CHU Sainte-Justine in Montreal (*n* = 86), British Columbia Children’s Hospital of Vancouver (*n* = 31), Winnipeg Children’s Hospital and CancerCare Manitoba (*n* = 28) and Alberta Children’s Hospital (*n* = 11). Patients were recruited approximately one month before transplant and followed up to one-year post-transplant for a maximum follow-up time of 13 months. Recruitment and follow-up began in July 2013 and ended in March 2017. At entry, a case report form (CRF) documented demographic data and pre-transplant clinical indicators such as age, sex, primary diagnosis, previous chemotherapy, conditioning regimen, graft source, donor type (matched or mismatched), EBV serology, and antiviral prophylaxis. During follow-up, data were collected prospectively through CRFs for variables related to diagnosed infections and treatment received. HHV DNAemia was diagnosed following confirmed positive testing by polymerase chain reaction (PCR). Appendix A provides details on the EBV and CMV qPCR tests used in each study site. Completed CRFs were sent prospectively to the coordination center at Sainte-Justine Hospital and data were included in the ACCESS database. This study protocol was approved by the research ethics boards of all four participating institutions, with patient consent waived as all data used in this analysis were collected through medical charts.

### 2.2. Statistical Analysis

Cumulative incidence (and 95% confidence interval (CI)) of EBV and CMV DNAemia were estimated using the Kaplan–Meier method and compared according to the antivirals (acyclovir and famciclovir) using log-rank testing. The curve for EBV DNAemia was stratified according to the pre-transplant EBV serological status of recipients. Follow-up time was considered from date of transplant up to the date of the EBV or CMV DNAemia event or, for censored information, up to 100 days post-transplant. A proportional hazard Cox regression model was used to measure the association between EBV and CMV DNAemia and antiviral use (famciclovir or acyclovir). Adjusted hazard ratio (HR) and 95% CI were estimated. Confounding was empirically controlled using the 5% change in estimate method for the following potential variables: age (continuous), primary diagnosis (malignant or non-malignant), graft source (stem cell peripheral blood/bone marrow or cord blood), conditioning regimen (myeloablative conditioning or other conditioning), sex (female or male), donor match (alternative or matched related donor), recipient pre-transplant EBV serology (negative, positive or unknown), graft donor EBV serostatus (negative, positive or unknown), Graft-versus-host disease (GvHD) (yes or no), and for the use (yes or no) of antithymocyte globulin (ATG), alemtuzumab, tacrolimus (FK506) or cyclosporine A (CsA), methotrexate (MTX), and mycophenolate mofetil (MMF). Multivariate models included the above-mentioned variables that changed the HR by ±5%. All analyses were done using STATA statistical software, version 14.2 (StataCorp, College Station, TX, USA).

## 3. Results

Table 1 reports the characteristics of the 156 patients included in the TREASuRE study according to the occurrence of post-transplant HHV DNAemia diagnosed during follow-up. Of these, 79 (50.6%) had at least one active HHV episode throughout the follow-up period (up to day 100 post-transplant) including 53 (34%) EBV episodes and 31 (19.9%) CMV episodes. Only a few cases of HSV (*n* = 3), VZV (*n* = 1) and HHV-6 (*n* = 4) were diagnosed. The post-transplant cumulative incidences of HSV, VZV, EBV, CMV, and HHV-6 DNAemia after 100 days of follow-up were, respectively, 2.5% (95% CI 0.8–7.6), 0.9% (95% CI: 0.1–6.1), 34.5% (95% CI: 27.6–42.6), 19.9% (95% CI: 14.5–27.1), and 3.4% (95% CI: 1.2–9.1). The mean age at transplant for all patients included in the analysis was 7.3 years (standard deviation (SD) ± 5.3). There were 117 patients (75%) treated with intravenous acyclovir and 43 patients (27.6%) with famciclovir. No other antiviral was given to recipients except for four patients who received ganciclovir (two who also received acyclovir and two who received famciclovir). Appendix A provides the characteristics of subjects according to antiviral use.

Figure 1 and Figure 2 illustrate the cumulative incidence of CMV and EBV DNAemia in transplant recipients according to antiviral treatment. Table 2 shows the estimates for associations between antivirals (acyclovir or famciclovir) and post-transplant EBV and CMV DNAemia. There was no significant difference between patients on antiviral treatment versus untreated. The adjusted HR for the relationship between acyclovir and EBV and CMV DNAemia were respectively 1.41 (95% CI: 0.63–3.14) and 0.55 (95% CI: 0.24–1.26). For famciclovir, the adjusted HR for EBV and CMV DNAemia were, respectively, 0.80 (95% CI: 0.42–1.51) and 0.82 (95% CI: 0.30–2.29).

## 4. Discussion

In this study, we found no statistically significant protective effect of acyclovir or famciclovir prophylaxis on EBV and CMV DNAemia incidence in our HSCT pediatric population. Other herpesvirus infections were infrequent with less than 5% cumulative incidence. Only three patients had HSV infection within 100 days post-transplant. The efficacy of acyclovir in the prophylaxis of HSV among seropositive patients has been demonstrated previously [20,25,26].

There was only one case of VZV infection in our cohort of 156 HSCT patients. The efficacy of acyclovir in the prevention of VZV reactivation has also been shown in numerous prior studies [20,24,25,26,27,28,29,30,31,32,33]. Moreover, universal vaccination against VZV with an attenuated wild strain, recommended since 1999 in all Canadian provinces [34], has also likely reduced wild-type VZV exposure in our cohort.

CMV reactivation is common after allogeneic HSCT [32]. In our cohort, the cumulative incidence of CMV DNAemia at 100 days post-transplant was 19.9% (95% CI: 14.5–27.1), while it ranged from 21.8% to 24% within four months in similar pediatric HSCT studies [6,35,36]. Importantly, in our study, the incidence of CMV DNAemia remained unchanged with the use of acyclovir or famciclovir. This is similar to findings reported by Selby et al. [37], Lundgren et al. [38], Ljungman et al. [39], and Prentice et al. [40] who suggested that acyclovir was ineffective for the prevention of CMV infection. However, some studies have shown that a high dose of acyclovir might have a protective effect against CMV among HSCT recipients. Prentice et al. [41] undertook a randomized study to compare the long-term (one-year) efficacy of acyclovir in three groups: group A (intravenous acyclovir 500 mg/m^2^ given three times/day from day 3 to day 30, then oral acyclovir 800 mg four times/day from day 31 to 210); group B (intravenous acyclovir 500 mg/m^2^, three times/day from day 3 to day 30, then placebo from day 31 to day 210); and group C (oral acyclovir 400 mg, four times/day from day 3 to day 30, then placebo from day 31 to day 210). Their survival analysis showed a 360-day post-transplant cumulative incidence of CMV viremia of 54% in group A, 50% in group B and 60% in group C. The difference between group B and group C was statistically significant (*p* = 0.03). Meyers et al. [42] studied the efficacy of acyclovir in a cohort of CMV seropositive HSCT recipients and showed that acyclovir significantly reduced the probability of CMV infection (0.70% vs. 0.87% *p* = 0.0001) and CMV disease (22% vs. 38% *p* = 0.008) after 100 days of follow-up among the group of patients who received intravenous acyclovir 500 mg/m^2^ every 8 h from day 5 to day 30 post-transplant, in comparison to the group which did not receive acyclovir. Another randomized study by Gluckman et al. [43] on the prophylaxis of herpesviruses with oral acyclovir 200 mg every 6 h from day 8 to day 35 post-transplant reported that oral acyclovir was effective against CMV compared to placebo on day 35 (0% vs. 7% *p* ˂ 0.007). It appears from these studies that the use of high-dose acyclovir may have a certain effect on CMV. However, despite a possible protective effect noted in some studies, the number of incident cases of CMV remains high in the exposed groups. Indeed, at the dose commonly prescribed as prophylaxis for HSV or VZV seropositive recipients, acyclovir does not appear to influence the incidence of CMV. Our study also showed no potential impact of famciclovir on CMV. The effect of famciclovir on CMV has been examined in only one study among adults [44], which showed that CMV reactivation was higher in seropositive patients who received oral acyclovir or famciclovir (*p* = 0.0001) compared to those who received oral valacyclovir and ganciclovir.

In our cohort, the 100-day post-transplant cumulative incidence of EBV DNAemia was 34.5% (95% CI: 27.6–42.6). This is similar to the cumulative incidence reported in several studies ranging from 22.6% to 32% in a follow-up period of one to two years among pediatric HSCT recipients [45,46,47]. Our study showed no effect of acyclovir or famciclovir on the incidence of EBV. Hann et al. [48] also found no reduction in the risk of EBV with the use of acyclovir in a randomized study. Similarly, Paula et al. [49], Krzysztof et al. [50], and Zutter et al. [51] showed no effect of acyclovir on EBV incidence. To our knowledge, no study on famciclovir and EBV has been conducted among pediatric HSCT recipients. However, among children with solid organ transplants, a meta-analysis [52] showed no significant effect of famciclovir on EBV (adjusted HR = 0.80 (95% CI: 0.42–1.51)).

These results are consistent with the mechanism of action of acyclovir-based anti-herpesvirus drugs. In their active form, acyclovir and famciclovir are deoxyguanosine analogs that competitively inhibit herpesvirus DNA polymerase causing arrest of lytic-cycle viral DNA replication [53]. These drugs have no effect on latent EBV DNA replication which is dependant on the cellular polymerase, thereby explaining their inefficacy in our study and in other studies referenced above. In the case of CMV, its lack of a specific thymidine kinase [53], which converts these drugs to their active form, would explain the observed lack of effect in clinical trials.

Our study has several strengths and limitations. It is a multicenter study with a good sample size and external validity. Although we cannot rule out a lack of power in our results, our cohort is one of the largest prospective studies on EBV outcomes in pediatric HSCT. However, as with any observational study, residual confounding cannot be excluded, although we attempted to compensate with rigorous and meticulous adjustment. Unmeasured variables such as pre-transplant CMV serostatus may also have led to residual confounding. Furthermore, our study focused on HHV measured by qPCR and did not distinguish between DNAemia and disease. This must be interpreted accordingly. It is also difficult to study the effect of different antiviral doses in an observational cohort study. Exposure to antivirals was analyzed by considering standard doses administered according to clinical guidelines and monitoring in pediatric HSCT programs. While beneficial effects of higher acyclovir and famciclovir doses on EBV and CMV may be possible, future clinical studies should target the effect of other more novel antivirals (such as maribavir) in pediatric populations. Maribavir is still pending approval (with its last phase III study completed in August 2020), but may provide some antiviral effect on EBV and CMV [54]. Letermovir is a new CMV inhibitor that targets the viral terminase complex to disrupt CMV DNA packaging (it was approved by the FDA in November 2017 and by Health Canada in June 2018 for prevention of clinically significant CMV infection in adult recipients of an allogeneic stem cell transplant in cases of contraindication or resistance to other antivirals for prophylaxis against CMV) [55]. Neither of these antivirals are known to impact HSV or VZV, but there are reports of inhibitory activity against EBV and/or CMV [54,55]. However, neither maribavir (still pending approval) nor letermovir (approved for CMV prophylaxis in adults only) was given to patients included in our study. In addition, there are several other potentially more promising avenues (such as cytotoxic T lymphocyte (CTL) therapy) that require further study.

## 5. Conclusions

Our study suggests a lack of efficacy of antiviral prophylaxis with acyclovir and famciclovir on EBV and CMV DNAemia in a large cohort of pediatric HSCT recipients. Prevention of active herpesvirus infections that can cause severe morbidity in HSCT recipients continues to be crucial to the success of the transplant. Further studies are needed to better delineate the potential impact of several other novel therapies that could have a more significant effect on these viruses.

## Figures and Tables

**Figure 1 vaccines-09-00610-f001:**
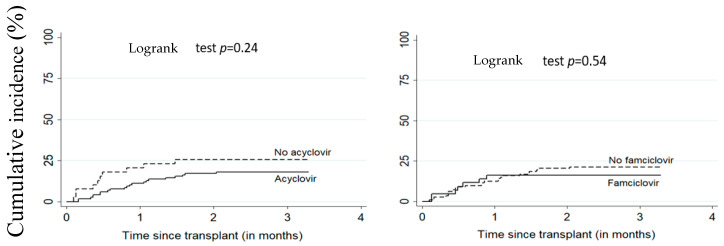
Cumulative incidence of CMV DNAemia according to acyclovir or famciclovir use. All patients (*n* = 156), CMV: cytomegalovirus.

**Figure 2 vaccines-09-00610-f002:**
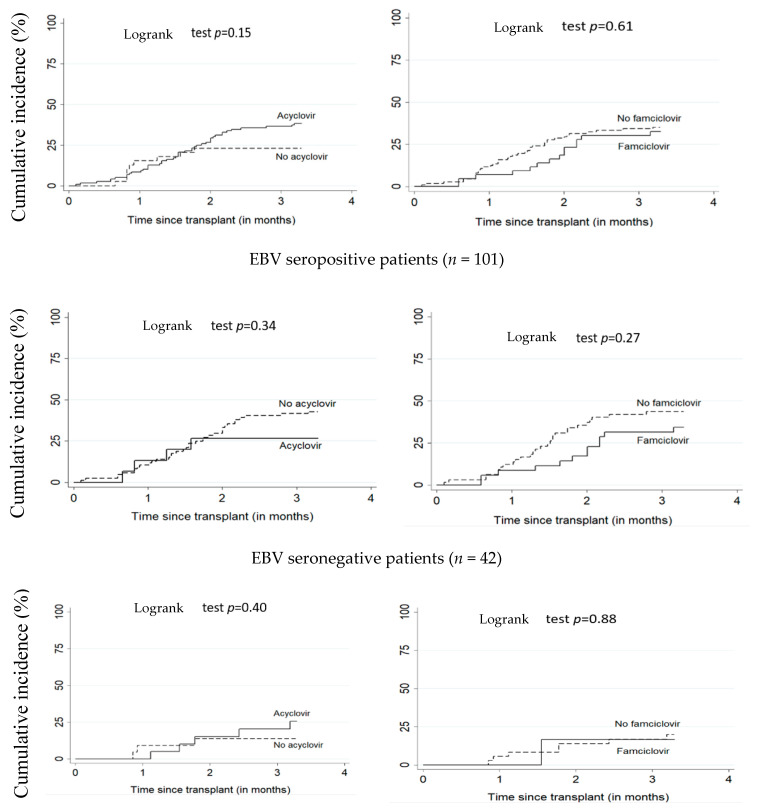
Cumulative incidence of EBV DNAemia according to acyclovir and famciclovir use. All patients (*n* = 156). EBV: Epstein–Barr virus.

**Table 1 vaccines-09-00610-t001:** Characteristics of HSCT recipients according to post-transplant HHV DNAemia.

Variables		Overall	HSV-1+	VZV+	EBV+	CMV+	HHV-6+	Total Positive HHV	Total Negative HHV
Number of Patients (*n*)		*n* = 156	*n* = 3	*n* = 1	*n* = 53	*n* = 31	*n* = 4	*n* = 77	*n* = 79
Sex, *n* (%)	Male	83 (53.2)	2(66.7)	0	26 (49.1)	14 (45.2)	3 (75.0)	44 (57.1)	39 (49.4)
	Female	73(46.8)	1(33.3)	1 (100)	27 (50.9)	17 (54.8)	1 (25.0)	33 (42.9)	40 (50.6)
Recipient age at transplant (years)	Mean (SD)	7.3 (5.3)	4.0 (2.5)	6.8	8.0 (5.6)	6.8 (4.9)	4.9 (2.8)	7.3 (5.4)	7.3 (5.3)
	Median (IQR)	6.3 (2.5,10.4)	5.2 (1.2,5.6)	6.8	7.3 (3.8, 10.6)	5.8 (2.3, 9.6)	4.9 (2.7, 7.1)	6.1 (2.4, 11.2)	6.5 (2.9, 10.1)
Primary diagnosis, *n* (%)	Malignant	69 (44.2)	1(33.3)	1 (100)	21 (39.6)	12 (38.7)	2 (50.0)	35 (45.5)	34 (43)
	Non-malignant	87 (55.8)	2(66.7)	0	32 (60.4)	19 (61.3)	2 (50.0)	42 (54.5)	45 (57)
Recipient pre-transplant EBV serology, *n* (%)	Negative	42 (26.9)	0	0	8 (15.1)	8 (25.8)	1 (25.0)	26 (33.8)	16 (20.3)
	Positive	101(64.8)	3(100)	1 (100)	40 (75.5)	21 (67.7)	3 (75.0)	43 (55.8)	58 (73.4)
	Unknown	13 (8.3)	0	0	5 (9.4)	2 (6.5)	0	8 (10.4)	5 (6.3)
Graft EBV serostatus, *n* (%)	Negative	62 (39.7)	2 (66.7)	1 (100)	8 (15.1)	12 (38.7)	3 (75.0)	39 (50.6)	23 (29.1)
	Positive	63 (40.4)	1 (33.3)	0	32 (60.4)	15 (48.4)	1 (25.0)	20 (26)	43 (54.4)
	Unknown	31 (19.9)	0	0	13 (24.5)	4 (12.9)	0	18 (23.4)	13 (16.5)
Donor match, *n* (%)	Donor matched	53 (34.0)	0	1 (100)	18 (34.0)	14 (45.2)	0	28 (36.4)	25 (31.6)
	Alternative donor	103 (66.0)	3 (100)	0	35 (66.0)	17 (54.8)	4 (100)	49 (63.6)	54 (68.4)
Graft source, *n* (%)	CB	39 (25.0)	2 (66.7)	0	5 (9.4)	9 (29.0)	3 (75.0)	23 (29.9)	16 (20.3)
	BM/PBSC	117 (75.0)	1 (33.3)	1 (100)	48 (90.6)	22 (71.0)	1 (25.0)	54 (70.1)	63 (79.7)
GvHD, *n* (%)	No	97 (62.18)	1 (33.33)	1 (100)	34 (64.15)	23 (74.19)	0	51 (66.23)	46 (58.23)
	Yes	59 (37.82)	2 (66.67)	0	19 (35.85)	8 (25.81)	4 (100)	26 (33.77)	33 (41.77)
Conditioning regimen, *n* (%)	Other	98 (62.8)	2 (66.7)	0	35 (66.0)	18 (58.1)	1 (25.0)	50 (64.9)	48 (60.8)
	MAC	58 (37.2)	1 (33.3)	1 (100)	18 (34.0)	13 (41.9)	3 (75.0)	27 (35.1)	31 (39.2)
Antithymocyte globulin, *n* (%)	No	92 (59.0)	3 (100)	1 (100)	19 (35.8)	19 (61.3)	3 (75.0)	52 (67.5)	40 (50.6)
	Yes	64 (41.0)	0	0	34 (64.2)	12 (38.7)	1 (25.0)	25 (32.5)	39 (49.4)
Alemtuzumab, *n* (%)	No	118(75.6)	1 (33.3)	1 (100)	44 (83.0)	24 (77.4)	3 (75.0)	55 (71.4)	63 (79.7)
	Yes	38 (24.4)	2 (66.7)	0	9 (17.0)	7 (22.6)	1 (25.0)	22 (28.6)	16 (20.3)
Tacrolimus or CsA, *n* (%)	No	15 (9.6)	0	0	3 (5.7)	3 (9.7)	2 (50.0)	8 (10.4)	7 (8.9)
	Yes	141(90.4)	3 (100)	1 (100)	50 (94.3)	28 (90.3)	2 (50.0)	69 (89.6)	72 (91.1)
MTX, *n* (%)	No	92 (59.0)	1 (33.3)	0	23 (43.4)	18 (58.1)	3 (75.0)	52 (67.5)	40 (50.6)
	Yes	64 (41.0)	2 (66.7)	1 (100)	30 (56.6)	13 (41.9)	1 (25.0)	25 (32.5)	39 (49.4)
MMF, *n* (%)	No	100(64.1)	2 (66.7)	1 (100)	46 (86.8)	21 (67.7)	2 (50.0)	39 (50.6)	61 (77.2)
	Yes	56 (35.9)	1 (33.3)	0	7 (13.2)	10 (32.3)	2 (50.0)	38 (49.4)	18 (22.7)
Acyclovir, *n* (%)	No	39 (25.0)	0	1 (100)	9 (17.0)	10 (32.3)	0	21 (27.3)	18 (22.8)
	Yes	117(75.0)	3 (100)	0	44 (83.0)	21 (67.7)	4 (100)	56 (72.7)	61 (77.2)
Famciclovir, *n* (%)	No	113(72.4)	2 (66.7)	1 (100)	39 (73.6)	24 (77.4)	4 (100)	54 (70.1)	59 (74.7)
	Yes	43 (27.6)	1 (33.3)	0	14 (26.4)	7 (22.6)	0	23 (29.9)	20 (25.3)
Other antivirals (Ganciclovir) *n* (%)	No	152 (97.44)	2 (66.67)	1 (100)	52 (98.11)	28 (90.32)	4 (100)	75 (94.94)	77 (100)
	Yes	4 (2.56)	1 (33.33)	0	1 (1.89)	3 (9.68)	0	4 (5.06)	0

ATG: antithymocyte globulin; BM: bone marrow; CB: cord blood; CsA: cyclosporine A; CMV: cytomegalovirus; EBV: Epstein–Barr virus; GvHD: Graft-versus-host disease; HHV: human herpesvirus; HHV-6: human herpesvirus 6; HSV-1: herpes simplex virus 1; HSCT: hematopoietic stem cell transplant; IQR: interquartile range; MAC: myeloablative conditioning; MMF: mycophenolate mofetil; MTX: methotrexate; NA: not applicable; PBSC: peripheral blood stem cells; SD: standard deviation; VZV: varicella-zoster virus.

**Table 2 vaccines-09-00610-t002:** Hazard ratios for the associations between antiviral prophylaxis (acyclovir or famciclovir) and post-transplant EBV and CMV DNAemia.

Variable		Number of Cases	Person-Time (Months)	Incidence Rate (95% CI)	HR Crude (95% CI)	HR Adjusted (95% CI)
EBV DNAemia		53	406.87	0.13 (0.1–0.17)		
Acyclovir ^(a)^	No	9	106.28	0.08 (0.04–0.16)	1	1
	Yes	44	300.58	0.15 (0.11–0.20)	1.68 (0.82–3.44)	1.41 (0.63–3.14)
Famciclovir ^(b)^	No	39	287.61	0.14 (0.10–0.19)	1	1
	Yes	14	119.26	0.12 (0.07–0.20)	0.85 (0.46–1.58)	0.79 (0.36–1.72)
CMV DNAemia		31	425.95	0.07 (0.05–0.10)		
Acyclovir ^(c)^	No	10	100.47	0.10 (0.05–0.18)	1	1
	Yes	21	325.49	0.06 (0.04–0.10)	0.64 (0.30–1.36)	0.55 (0.24–1.26)
Famciclovir ^(d)^	No	24	304.23	0.08 (0.05–0.12)	1	1
	Yes	7	121.72	0.06 (0.03–0.12)	0.77 (0.33–1.78)	0.82 (0.30–2.29)

CMV: cytomegalovirus; EBV: Epstein–Barr virus; HR: hazard ratio; CI: Confidence intervals. ^(a)^ The following variables were considered to estimate the adjusted hazard ratio: recipient age at transplant (continuous), recipient pre-transplant EBV serology (negative, positive or unknown), graft donor EBV serostatus (negative, positive or unknown), antithymocyte globulin (yes or no) and site of study. ^(b)^ The following variables were considered to estimate the adjusted hazard ratio: recipient age at transplant (continuous), recipient pre-transplant EBV serology (negative, positive or unknown), graft source (bone marrow/peripheral blood stem cells or cord blood), conditioning regimen (myeloablative or other), anti-thymocyte globulin (yes or no), mycophenolate mofetil (yes or no) and site of study. ^(c)^ The following variables were considered to estimate the adjusted hazard ratio: recipient sex (female or male) and recipient pre-transplant EBV serology (negative, positive, or unknown). ^(d)^ The following variables were considered to estimate adjusted hazard ratio: graft donor EBV serostatus (negative, positive or unknown), donor match (alternative or matched related donor), conditioning regimen (myeloablative or other) and site of study.

## Data Availability

The data presented in this study are available on request from the corresponding author. The data are not publicly available due to privacy/ethical restrictions.

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
