# Peer review of "Association between Antiviral Prophylaxis and Cytomegalovirus and Epstein–Barr Virus DNAemia in Pediatric Recipients of Allogeneic Hematopoietic Stem Cell Transplant"

_vaccines, 2021, doi:10.3390/vaccines9060610_

Round 1

Reviewer 1 Report

The article “Association between antiviral prophylaxis and cytomegalovirus and Epstein-Barr virus DNAemia in pediatric recipients of allogeneic hematopoietic stem cell transplant” by Diop et al analyzed the outcome of antiviral prophylaxis in CMV and EBV DNAemia in pediatric recipients of allogeneic hematopoietic stem cell transplant (HSCT) and observed insignificant protective effect of acyclovir and Famciclovir prophylaxis on EBV and CMV DNAemia incidence. The study was made on 156 HSCT patients’ data from four Canadian pediatric tertiary care centers. However, this study has major limitations and trivial contributions to the existing knowledge to the field of prophylaxis in CMV and EBV. The prophylaxis of acyclovir and the related drug has been studied extensively and the current study also supports similar outcomes in pediatric HSCT patients. The major drawback of these nucleoside analogs kind of drugs is well known and found to be ineffective in case of latently infected or actively proliferating cells with EBV infection and none of them received approval from FDA (Food and Drug Administration). Several literature reviews and articles are already available highlighting the effect of acyclovir prophylaxis and discussed the limitations of these drugs. So, the present article by Diop et al to failed to show any new findings or scientific contribution to the existing prophylaxis research field. This article also lacks the discussion of mechanisms behind resistance and limitations with traditional antiviral drugs and recent clinical studies with new drugs and their strategies in having good outcomes in other transplant patients.  

Author Response

Reviewer comments

We sincerely thank the reviewers for their constructive comments which have enabled us to significantly improve our manuscript.

Reviewer 1:

The article “Association between antiviral prophylaxis and cytomegalovirus and Epstein-Barr virus DNAemia in pediatric recipients of allogeneic hematopoietic stem cell transplant” by Diop et al analyzed the outcome of antiviral prophylaxis in CMV and EBV DNAemia in pediatric recipients of allogeneic hematopoietic stem cell transplant (HSCT) and observed insignificant protective effect of acyclovir and Famciclovir prophylaxis on EBV and CMV DNAemia incidence. The study was made on 156 HSCT patients’ data from four Canadian pediatric tertiary care centers. However, this study has major limitations and trivial contributions to the existing knowledge to the field of prophylaxis in CMV and EBV. The prophylaxis of acyclovir and the related drug has been studied extensively and the current study also supports similar outcomes in pediatric HSCT patients. The major drawback of these nucleoside analogs kind of drugs is well known and found to be ineffective in case of latently infected or actively proliferating cells with EBV infection and none of them received approval from FDA (Food and Drug Administration). Several literature reviews and articles are already available highlighting the effect of acyclovir prophylaxis and discussed the limitations of these drugs. So, the present article by Diop et al to failed to show any new findings or scientific contribution to the existing prophylaxis research field. This article also lacks the discussion of mechanisms behind resistance and limitations with traditional antiviral drugs and recent clinical studies with new drugs and their strategies in having good outcomes in other transplant patients.  

Response:

1) We agree with Reviewer 1 that this question has been addressed by other studies in the past, however the problem is that the data from such studies show mixed results. Also, there are few pediatric cohort studies in the existing literature. For these reasons we believe that our study not only provides relevant data but also discusses this data thoroughly in the context of past publications. We have added a sentence in the abstract (lines 28-29), and in the text (line 90-91) to highlight the importance of this study in the pediatric HSCT field. We also added the keyword “pediatric” in the list of keywords (line 45).

2) We thank the reviewer for comments on existing nucleoside analog-based anti-herpes drugs and potential new drugs in clinical development. We added two paragraphs in the Discussion section to discuss resistance mechanisms and limitations with traditional antiviral drugs (lines 246-253) and the development of new antivirals (lines 266-273). It is important to mention that there are no new anti-herpesvirus drugs in development for use in pediatric patients (clinicaltrials.gov). Among known drugs, there are: 1) letermovir (approved by Health Canada since 06/20/2018 for prevention of CMV in adult allogeneic HSCT patients), and 2) maribavir (which is still pending approval with last phase III study completed in August 2020). However, neither of these two antivirals have an effect on HSV or VZV, and they are not used in pediatrics (except for adolescents >16 years of age with adult weight and contraindications/ resistance to other antivirals for prophylaxis against CMV in the transplant setting, which happened once in our center in 2020).  

Reviewer 2 Report

This study addresses the question of whether use of acyclovir or famciclovir prophylaxis decreases the risk of CMV and EBV DNAemia among recipients (ages <21) of allogenic hematopoietic stem cell transplantation. This is a worthy question that has been addressed by other studies in the past with mixed results and, as the authors point out in the discussion, the answer may partly depend on the acyclovir dose applied.

For this report, the authors have analyzed data previously collected as part of a prospective observational study, entitled Transfusion Related-EBV Infection among Allogenic Stem Cell Pediatric Recipients (TREASuRE). They cite a previous publication that contains information about the TREASuRE study design and methods, but the information in that publication and in this manuscript is limited. The prespecified outcome measures for the TREASuRE study that are listed on ClinicalTrials.gov (NCT 0250789) differ from the those addressed in this report.

The authors indicate that standard of care includes use of ganciclovir prophylaxis in allogenic HSCT recipients at high risk of CMV infection. Ganciclovir/valganciclovir also has activity against HSV, VZV, and EBV. The authors do not report whether such patients were excluded from their study or if the preemptive approach to CMV disease prevention was applied to all their patients.

Of the 156 patients evaluated in the TREASuRE study, 75% had received acyclovir prophylaxis and 27.6% received famciclovir prophylaxis. The interpretation of the results requires knowing if participants in the no acyclovir or no famciclovir comparator arms received anti-herpesvirus drugs of any kind. These details should be provided.

Recommend including a table comparing the various characteristics of the no acyclovir group versus the acyclovir prophylaxis group.  

Variables of acute GVHD and CMV status recipient/donor (factors influencing CMV reactivation/infection risk) should be included in the analyses.

The methods state that follow-up data elements were collected. What type of data elements were collected? What was the frequency of missing data elements? 

What proportion of patients who received acyclovir or famciclovir prophylaxis continuously took the drug for the duration of 100 days post-transplant? for 6 weeks?

Line 99. RE: “Patients were recruited…”.  The recruitment process involving research participants implies that information about the study is provided to prospective subjects. Was this the case for the TREASuRE study? 

Author Response

Reviewer 2:

This study addresses the question of whether use of acyclovir or famciclovir prophylaxis decreases the risk of CMV and EBV DNAemia among recipients (ages <21) of allogenic hematopoietic stem cell transplantation. This is a worthy question that has been addressed by other studies in the past with mixed results and, as the authors point out in the discussion, the answer may partly depend on the acyclovir dose applied.

            Response: Thank you for this comment.

For this report, the authors have analyzed data previously collected as part of a prospective observational study, entitled Transfusion Related-EBV Infection among Allogenic Stem Cell Pediatric Recipients (TREASuRE). They cite a previous publication that contains information about the TREASuRE study design and methods, but the information in that publication and in this manuscript is limited. The prespecified outcome measures for the TREASuRE study that are listed on ClinicalTrials.gov (NCT 0250789) differ from the those addressed in this report.

Response: The TREASuRE study has several objectives. The primary objective was to determine the link between EBV infection among HSCT and blood product transfusion (published recently: Enok Bonong et al, Transfusion. 2021 Jan;61(1):144-158). The study also included secondary objectives among which was an analysis of the association between antivirals and EBV/CMV infections).

The authors indicate that standard of care includes use of ganciclovir prophylaxis in allogenic HSCT recipients at high risk of CMV infection. Ganciclovir/valganciclovir also has activity against HSV, VZV, and EBV. The authors do not report whether such patients were excluded from their study or if the preemptive approach to CMV disease prevention was applied to all their patients.

Response: No patient was excluded from our study. Only four patients received ganciclovir for CMV (two of whom received acyclovir and two received famciclovir). These data have been included in Table 1 (see track changes in Table 1).

Of the 156 patients evaluated in the TREASuRE study, 75% had received acyclovir prophylaxis and 27.6% received famciclovir prophylaxis. The interpretation of the results requires knowing if participants in the no acyclovir or no famciclovir comparator arms received anti-herpesvirus drugs of any kind. These details should be provided.

Response: No other antiviral was received by patients other than acyclovir and famciclovir except for the four patients who received ganciclovir. These data have been added in Table 1 and a sentence was added in the text (see lines: 148-150).

Recommend including a table comparing the various characteristics of the no acyclovir group versus the acyclovir prophylaxis group.  

Response: A supplementary Table S2 has been added as supplementary material providing characteristics according to antivirals. The reference to Table S2 is provided in the manuscript on lines 150-151 (see track changes).

Variables of acute GVHD and CMV status recipient/donor (factors influencing CMV reactivation/infection risk) should be included in the analyses.

Response: GvHD variable has been added in Table 1. GvHD was also considered as all other co-variables in the regression analysis models as described in the statistical analysis section (see addition using track changes, line 132).

The methods state that follow-up data elements were collected. What type of data elements were collected? What was the frequency of missing data elements? 

Response: Data collection was described from line 104 to line 114. There are no missing data. Completed data are presented in Table 1 without any missing data (except for the EBV serology status for which the proportion of missing data is described).

What proportion of patients who received acyclovir or famciclovir prophylaxis continuously took the drug for the duration of 100 days post-transplant? for 6 weeks?

Response: Antivirals were given according to the clinical recommendation. Therefore, all patients for whom antivirals were indicated were still taking them at day 100. Discontinuation is possible for toxicity reasons, but this is very rare (less than 1%).

Line 99. RE: “Patients were recruited…”.  The recruitment process involving research participants implies that information about the study is provided to prospective subjects. Was this the case for the TREASuRE study? 

Response: We apologize for the confusion. The terms « enrollment» and «recruitment » are used to signify that patients were identified based on their being enlisted in the transplant registry. Because all data were obtainable from medical charts as a result of standard-of-care tests, the ethics committees permitted institutional approval with waived patient consent.

Round 2

Reviewer 1 Report

The revised manuscript discussed the limitations of their result outcomes and highlighted the study as antiviral prophylaxis in pediatric patients. As per the suggestion, authors also discussed the promising alternate antivirals for the treatment. So, the revised manuscript by Diop et al  has addressed all the raised concerns and improved the overall presentation and quality of the study.

Author Response

Response: We sincerely thank the reviewer for the constructive comments which have enabled us to significantly improve our manuscript.

Reviewer 2 Report

The authors have done an excellent job responding to the reviewer’s comments and questions. I also like the changes in the manuscript emphasizing the scarcity of data in the pediatric population.

I have one remaining issue to settle:

Pre‐transplant CMV serostatus is one of the most important determinants of CMV reactivation after allogeneic HCST -- low risk (donor [D]−/recipient [R]−), intermediate risk (D+/R−), or high risk (D−/R+ or D+/R+).

The mean age of their population of 7 years (SD +/-5), compared to adults, will likely lower the proportion of allogenic HSCT recipients who were CMV-seropositive pretransplant and at higher risk of CMV reactivation/infection. CMV-serostatus is a potential confounder in the CMV outcome analysis of ACV/FCV vs no prophylaxis groups.

The authors did not address why the CMV serostatus information has not been included. A statement should be included in the methods section or discussion acknowledging the absence of pre‐transplant CMV serostatus information and its implication.

Author Response

Response: We thank the reviewer for this clarification. We have added a sentence in the Discussion section to address the important limitation regarding absence of pre-transplant CMV serostatus information (see lines 259-260, track changes).